# Projected Natural Actor-Critic

**Philip S. Thomas, William Dabney, Sridhar Mahadevan, and Stephen Giguere**
School of Computer Science
University of Massachusetts Amherst
Amherst, MA 01003
`{pthomas,wdabney,mahadeva,sgiguere}@cs.umass.edu`

## Abstract

Natural actor-critics form a popular class of policy search algorithms for finding locally optimal policies for Markov decision processes. In this paper we address a drawback of natural actor-critics that limits their real-world applicability—their lack of safety guarantees. We present a principled algorithm for performing natural gradient descent over a constrained domain. In the context of reinforcement learning, this allows for natural actor-critic algorithms that are guaranteed to remain within a known safe region of policy space. While deriving our class of constrained natural actor-critic algorithms, which we call Projected Natural Actor-Critics (PNACs), we also elucidate the relationship between natural gradient descent and mirror descent.

## 1 Introduction

Natural actor-critics form a class of policy search algorithms for finding locally optimal policies for Markov decision processes (MDPs) by approximating and ascending the natural gradient [1] of an objective function. Despite the numerous successes of, and the continually growing interest in, natural actor-critic algorithms, they have not achieved widespread use for real-world applications. A lack of safety guarantees is a common reason for avoiding the use of natural actor-critic algorithms, particularly for biomedical applications. Since natural actor-critics are *unconstrained* optimization algorithms, there are no guarantees that they will avoid regions of policy space that are known to be dangerous.

For example, proportional-integral-derivative controllers (PID controllers) are the most widely used control algorithms in industry, and have been studied in depth [2]. Techniques exist for determining the set of stable gains (policy parameters) when a model of the system is available [3]. Policy search can be used to find the optimal gains within this set (for some definition of optimality). A desirable property of a policy search algorithm in this context would be a guarantee that it will remain within the predicted region of stable gains during its search.

Consider a second example: *functional electrical stimulation* (FES) control of a human arm. By selectively stimulating muscles using subcutaneous probes, researchers have made significant strides toward returning motor control to people suffering from paralysis induced by spinal cord injury [4]. There has been a recent push to develop controllers that specify how much and when to stimulate each muscle in a human arm to move it from its current position to a desired position [5]. This closed-loop control problem is particularly challenging because each person's arm has different dynamics due to differences in, for example, length, mass, strength, clothing, and amounts of muscle atrophy, spasticity, and fatigue. Moreover, these differences are challenging to model. Hence, a proportional-derivative (PD) controller, tuned to a simulation of an ideal human arm, required manual tuning to obtain desirable performance on a human subject with biceps spasticity [6].

Researchers have shown that policy search algorithms are a viable approach to creating controllers that can automatically adapt to an individual's arm by training on a few hundred two-second reach-

ing movements [7]. However, safety concerns have been raised in regard to both this specific application and other biomedical applications of policy search algorithms. Specifically, the existing state-of-the-art gradient-based algorithms, including the current natural actor-critic algorithms, are unconstrained and could potentially select dangerous policies. For example, it is known that certain muscle stimulations could cause the dislocation of a subject's arm. Although we lack an accurate model of each individual's arm, we can generate conservative safety constraints on the space of policies. Once again, a desirable property of a policy search algorithm would be a guarantee that it will remain within a specified region of policy space (known-safe policies).

In this paper we present a class of natural actor-critic algorithms that perform constrained optimization—given a known safe region of policy space, they search for a locally optimal policy while always remaining within the specified region. We call our class of algorithms *Projected Natural Actor-Critics* (PNACs) since, whenever they generate a new policy, they project the policy back to the set of safe policies. The interesting question is how the projection can be done in a principled manner. We show that natural gradient descent (ascent), which is an *unconstrained* optimization algorithm, is a special case of mirror descent (ascent), which is a *constrained* optimization algorithm. In order to create a projected natural gradient algorithm, we add constraints in the mirror descent algorithm that is equivalent to natural gradient descent. We apply this projected natural gradient algorithm to policy search to create the PNAC algorithms, which we validate empirically.

## 2 Related Work

Researchers have addressed safety concerns like these before [8]. Bendrahim and Franklin [9] showed how a walking biped robot can switch to a stabilizing controller whenever the robot leaves a stable region of state space. Similar state-avoidant approaches to safety have been proposed by several others [10, 11, 12]. These approaches do not account for situations where, over an unavoidable region of state space, the actions themselves are dangerous. Kuindersma et al. [13] developed a method for performing risk-sensitive policy search, which models the variance of the objective function for each policy and permits runtime adjustments of risk sensitivity. However, their approach does not guarantee that an unsafe region of state space or policy space will be avoided.

Bhatnagar et al. [14] presented projected natural actor-critic algorithms for the average reward setting. As in our projected natural actor-critic algorithms, they proposed computing the update to the policy parameters and then projecting back to the set of allowed policy parameters. However, they did not specify how the projection could be done in a principled manner. We show in Section 7 that the Euclidean projection can be arbitrarily bad, and argue that the projection that we propose is particularly compatible with natural actor-critics (natural gradient descent).

Duchi et al. [15] presented mirror descent using the Mahalanobis norm for the proximal function, which is very similar to the proximal function that we show to cause mirror descent to be equivalent to natural gradient descent. However, their proximal function is not identical to ours and they did not discuss any possible relationship between mirror descent and natural gradient descent.

## 3 Natural Gradients

Consider the problem of minimizing a differentiable function $f : \mathbb{R}^n \to \mathbb{R}$. The standard gradient descent approach is to select an initial $x_0 \in \mathbb{R}^n$, compute the direction of steepest descent, $-\nabla f(x_0)$, and then move some amount in that direction (scaled by a step size parameter, $\alpha_0$). This process is then repeated indefinitely: $x_{k+1} = x_k - \alpha_k \nabla f(x_k)$, where $\{\alpha_k\}$ is a step size schedule and $k \in \{1, \ldots\}$. Gradient descent has been criticized for its low asymptotic rate of convergence. Natural gradients are a quasi-Newton approach to improving the convergence rate of gradient descent.

When computing the direction of steepest descent, gradient descent assumes that the vector $x_k$ resides in Euclidean space. However, in several settings it is more appropriate to assume that $x_k$ resides in a Riemannian space with metric tensor $G(x_k)$, which is an $n \times n$ positive definite matrix that may vary with $x_k$ [16]. In this case, the direction of steepest descent is called the *natural gradient* and is given by $-G(x_k)^{-1}\nabla f(x_k)$ [1]. In certain cases, (which include our policy search application), following the natural gradient is asymptotically Fisher-efficient [16].

# 4 Mirror Descent

Mirror descent algorithms form a class of highly scalable online gradient methods that are useful in constrained minimization of non-smooth functions [17, 18]. They have recently been applied to value function approximation and basis adaptation for reinforcement learning [19, 20]. The mirror descent update is

$$x_{k+1} = \nabla\psi_k^*\big(\nabla\psi_k(x_k) - \alpha_k\nabla f(x_k)\big), \tag{1}$$

where $\psi_k : \mathbb{R}^n \to \mathbb{R}$ is a continuously differentiable and strongly convex function called the *proximal function*, and where the conjugate of $\psi_k$ is $\psi_k^*(y) \triangleq \max_{x\in\mathbb{R}^n}\{x^\intercal y - \psi_k(x)\}$, for any $y \in \mathbb{R}^n$. Different choices of $\psi_k$ result in different mirror descent algorithms. A common choice for a fixed $\psi_k = \psi, \forall k$, is the $p$-norm [20], and a common adaptive $\psi_k$ is the Mahalanobis norm with a dynamic covariance matrix [15].

Intuitively, the distance metric for the space that $x_k$ resides in is not necessarily the same as that of the space that $\nabla f(x_k)$ resides in. This suggests that it may not be appropriate to directly add $x_k$ and $-\alpha_k\nabla f(x_k)$ in the gradient descent update. To correct this, mirror descent moves $x_k$ into the space of gradients (the *dual space*) with $\nabla\psi_k(x_k)$ before performing the gradient update. It takes the result of this step in gradient space and returns it to the space of $x_k$ (the *primal space*) with $\nabla\psi_k^*$. Different choices of $\psi_k$ amount to different assumptions about the relationship between the primal and dual spaces at $x_k$.

# 5 Equivalence of Natural Gradient Descent and Mirror Descent

**Theorem 5.1.** *The natural gradient descent update at step $k$ with metric tensor $G_k \triangleq G(x_k)$:*

$$x_{k+1} = x_k - \alpha_k G_k^{-1}\nabla f(x_k), \tag{2}$$

*is equivalent to* (1)*, the mirror descent update at step $k$, with $\psi_k(x) = (1/2)x^\intercal G_k x$.*

*Proof.* First, notice that $\nabla\psi_k(x) = G_k x$. Next, we derive a closed-form for $\psi_k^*$:

$$\psi_k^*(y) = \max_{x\in\mathbb{R}^n}\left\{x^\intercal y - \frac{1}{2}x^\intercal G_k x\right\}. \tag{3}$$

Since the function being maximized on the right hand side is strictly concave, the $x$ that maximizes it is its critical point. Solving for this critical point, we get $x = G_k^{-1}y$. Substituting this into (3), we find that $\psi_k^*(y) = (1/2)y^\intercal G_k^{-1}y$. Hence, $\nabla\psi_k^*(y) = G_k^{-1}y$. Inserting the definitions of $\nabla\psi_k(x)$ and $\nabla\psi_k^*(y)$ into (1), we find that the mirror descent update is

$$x_{k+1} = G_k^{-1}\left(G_k x_k - \alpha_k\nabla f(x_k)\right) = x_k - \alpha_k G_k^{-1}\nabla f(x_k),$$

which is identical to (2). ∎

Although researchers often use $\psi_k$ that are norms like the $p$-norm and Mahalanobis norm, notice that the $\psi_k$ that results in natural gradient descent is *not* a norm. Also, since $G_k$ depends on $k$, $\psi_k$ is an *adaptive* proximal function [15].

# 6 Projected Natural Gradients

When $x$ is constrained to some set, $X$, $\psi_k$ in mirror descent is augmented with the indicator function $I_X$, where $I_X(x) = 0$ if $x \in X$, and $+\infty$ otherwise. The $\psi_k$ that was shown to generate an update equivalent to the natural gradient descent update, with the added constraint that $x \in X$, is $\psi_k(x) = (1/2)x^\intercal G_k x + I_X(x)$. Hereafter, any references to $\psi_k$ refer to this augmented version.

For this proximal function, the subdifferential of $\psi_k(x)$ is $\nabla\psi_k(x) = G_k(x) + \hat{N}_X(x) = (G_k + \hat{N}_X)(x)$, where $\hat{N}_X(x) \triangleq \partial I_X(x)$ and, in the middle term, $G_k$ and $\hat{N}_X$ are relations and $+$ denotes Minkowski addition.[1] $\hat{N}_X(x)$ is the normal cone of $X$ at $x$ if $x \in X$ and $\emptyset$ otherwise [21].

$$\nabla\psi_k^*(y) = (G_k + \hat{N}_X)^{-1}(y). \tag{4}$$

Let $\Pi_X^{G_k}(y)$, be the set of $x \in X$ that are closest to $y$, where the length of a vector, $z$, is $(1/2)z^\mathsf{T}G_k z$. More formally,

$$\Pi_X^{G_k}(y) \triangleq \arg\min_{x \in X} \frac{1}{2}(y-x)^\mathsf{T}G_k(y-x). \tag{5}$$

**Lemma 6.1.** $\Pi_X^{G_k}(y) = (G_k + \hat{N}_X)^{-1}(G_k y)$.

*Proof.* We write (5) without the explicit constraint that $x \in X$ by appending the indicator function:

$$\Pi_X^{G_k}(y) = \arg\min_{x \in \mathbb{R}^n} h_y(x),$$

where $h_y(x) = (1/2)(y-x)^\mathsf{T}G_k(y-x) + I_X(x)$. Since $h_y$ is strictly convex over $X$ and $+\infty$ elsewhere, its critical point is its global minimizer. The critical point satisfies

$$0 \in \nabla h_y(x) = -G_k(y) + G_k(x) + \hat{N}_X(x).$$

The globally minimizing $x$ therefore satisfies $G_k y \in G_k(x) + \hat{N}_X(x) = (G_k + \hat{N}_X)(x)$. Solving for $x$, we find that $x = (G_k + \hat{N}_X)^{-1}(G_k y)$. ∎

Combining Lemma 6.1 with (4), we find that $\nabla \psi^*(y) = \Pi_X^{G_k}(G_k^{-1}y)$. Hence, mirror descent with the proximal function that produces natural gradient descent, augmented to include the constraint that $x \in X$, is:

$$\begin{aligned}
x_{k+1} &= \Pi_X^{G_k}\left(G_k^{-1}\left((G_k + \hat{N}_X)(x_k) - \alpha_k \nabla f(x_k)\right)\right) \\
&= \Pi_X^{G_k}\left((I + G_k^{-1}\hat{N}_X)(x_k) - \alpha_k G_k^{-1}\nabla f(x_k)\right),
\end{aligned}$$

where $I$ denotes the identity relation. Since $x_k \in X$, we know that $0 \in \hat{N}_X(x_k)$, and hence the update can be written as

$$x_{k+1} = \Pi_X^{G_k}\left(x_k - \alpha_k G_k^{-1}\nabla f(x_k)\right), \tag{6}$$

which we call *projected natural gradient* (PNG).

## 7 Compatibility of Projection

The standard projected subgradient (PSG) descent method follows the negative gradient (as opposed to the negative natural gradient) and projects back to $X$ using the Euclidean norm. If $f$ and $X$ are convex and the step size is decayed appropriately, it is guaranteed to converge to a global minimum, $x^* \in X$. Any such $x^*$ is a fixed point. This means that a small step in the negative direction of any subdifferential of $f$ at $x^*$ will project back to $x^*$.

Our choice of projection, $\Pi_X^{G_k}$, results in PNG having the same fixed points (see Lemma 7.1). This means that, when the algorithm is at $x^*$ and a small step is taken down the natural gradient to $x'$, $\Pi_X^{G_k}$ will project $x'$ back to $x^*$. We therefore say that $\Pi_X^{G_k}$ is compatible with the natural gradient. For comparison, the Euclidean projection of $x'$ will *not* necessarily return $x'$ to $x^*$.

**Lemma 7.1.** *The sets of fixed points for PSG and PNG are equivalent.*

*Proof.* A necessary and sufficient condition for $x$ to be a fixed point of PSG is that $-\nabla f(x) \in \hat{N}_X(x)$ [22]. A necessary and sufficient condition for $x$ to be a fixed point of PNG is

$$\begin{aligned}
x &= \Pi_X^{G_k}\left(x - \alpha_k G_k^{-1}\nabla f(x)\right) = (G_k + \hat{N}_X)^{-1}\left(G_k\left(x - \alpha_k G_k^{-1}\nabla f(x)\right)\right) \\
&= (G_k + \hat{N}_X)^{-1}\left(G_k x - \alpha_k \nabla f(x)\right) \\
&\Leftrightarrow G_k x - \alpha_k \nabla f(x) \in G_k(x) + \hat{N}_X(x) \\
&\Leftrightarrow -\nabla f(x) \in \hat{N}_X(x). \quad\blacksquare
\end{aligned}$$

To emphasize the importance of using a compatible projection, consider the following simple example. Minimize the function $f(x) = x^\mathsf{T}Ax + b^\mathsf{T}x$, where $A = \mathrm{diag}(1, 0.01)$ and $b = [-0.2, -0.1]^\mathsf{T}$, subject to the constraints $\|x\|_1 \leq 1$ and $x \geq 0$. We implemented three algorithms, and ran each for 1000 iterations using a fixed step size:

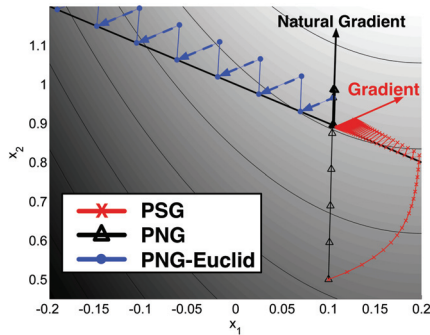

Figure 1: The thick diagonal line shows one constraint and dotted lines show projections. Solid arrows show the directions of the natural gradient and gradient at the optimal solution, $x^*$. The dashed blue arrows show PNG-Euclid's projections, and emphasize the the projections cause PNG-Euclid to move *away* from the optimal solution.

1. **PSG** - projected subgradient descent using the Euclidean projection.
2. **PNG** - projected natural gradient descent using $\Pi_X^{G_k}$.
3. **PNG-Euclid** - projected natural gradient descent using the Euclidean projection.

The results are shown in Figure 1. Notice that PNG and PSG converge to the optimal solution, $x^*$. From this point, they both step in different directions, but project back to $x^*$. However, PNG-Euclid converges to a suboptimal solution (outside the domain of the figure). If $X$ were a line segment between the point that PNG-Euclid and PNG converge to, then PNG-Euclid would converge to the pessimal solution within $X$, while PSG and PNG would converge to the optimal solution within $X$. Also, notice that the natural gradient corrects for the curvature of the function and heads directly towards the global unconstrained minimum. Since the natural methods in this example use metric tensor $G = A$, which is the Hessian of $f$, they are essentially an incremental form of Newton's method. In practice, the Hessian is usually not known, and an estimate thereof is used.

## 8 Natural Actor-Critic Algorithms

An MDP is a tuple $M = (\mathcal{S}, \mathcal{A}, \mathcal{P}, \mathcal{R}, d_0, \gamma)$, where $\mathcal{S}$ is a set of states, $\mathcal{A}$ is a set of actions, $\mathcal{P}(s'|s,a)$ gives the probability density of the system entering state $s'$ when action $a$ is taken in state $s$, $R(s,a)$ is the expected reward, $r$, when action $a$ is taken in state $s$, $d_0$ is the initial state distribution, and $\gamma \in [0,1)$ is a reward discount parameter. A parameterized policy, $\pi$, is a conditional probability density function—$\pi(a|s,\theta)$ is the probability density of action $a$ in state $s$ given a vector of policy parameters, $\theta \in \mathbb{R}^n$.

Let $J(\theta) = \mathrm{E}\left[\sum_{t=0}^{\infty} \gamma^t r_t | \theta\right]$ be the *discounted-reward objective* or the *average reward objective function* with $J(\theta) = \lim_{n \to \infty} \frac{1}{n} \mathrm{E}\left[\sum_{t=0}^{n} r_t | \theta\right]$. Given an MDP, $M$, and a parameterized policy, $\pi$, the goal is to find policy parameters that maximize one of these objectives. When the action set is continuous, the search for globally optimal policy parameters becomes intractable, so policy search algorithms typically search for locally optimal policy parameters.

Natural actor-critics, first proposed by Kakade [23], are algorithms that estimate and ascend the natural gradient of $J(\theta)$, using the average Fisher information matrix as the metric tensor:

$$G_k = G(\theta_k) = \mathrm{E}_{s \sim d^\pi, a \sim \pi}\left[\left(\frac{\partial}{\partial \theta_k} \log \pi(a|s,\theta_k)\right)\left(\frac{\partial}{\partial \theta_k} \log \pi(a|s,\theta_k)\right)^\mathsf{T}\right],$$

where $d^\pi$ is a policy and objective function-dependent distribution over the state set [24].

There are many natural actor-critics, including Natural policy gradient utilizing the Temporal Differences (NTD) algorithm [25], Natural Actor-Critic using LSTD-Q($\lambda$) (NAC-LSTD) [26], Episodic Natural Actor-Critic (eNAC) [26], Natural Actor-Critic using Sarsa($\lambda$) (NAC-Sarsa) [27], Incremental Natural Actor-Critic (INAC) [28], and Natural-Gradient Actor-Critic with Advantage Parameters (NGAC) [14]. All of them form an estimate, typically denoted $w_k$, of the natural gradient of $J(\theta_k)$. That is, $w_k \approx G(\theta_k)^{-1} \nabla J(\theta_k)$. They then perform the policy parameter update, $\theta_{k+1} = \theta_k + \alpha_k w_k$.

## 9 Projected Natural Actor-Critics

If we are given a closed convex set, $\Theta \subseteq \mathbb{R}^n$, of admissible policy parameters (e.g., the stable region of gains for a PID controller), we may wish to ensure that the policy parameters remain

within $\Theta$. The natural actor-critic algorithms described in the previous section do not provide such a guarantee. However, their policy parameter update equations, which are natural gradient ascent updates, can easily be modified to the projected natural gradient ascent update in (6) by projecting the parameters back onto $\Theta$ using $\Pi_\Theta^{G(\theta_k)}$:

$$\theta_{k+1} = \Pi_\Theta^{G(\theta_k)}(\theta_k + \alpha_k w_k).$$

Many of the existing natural policy gradient algorithms, including NAC-LSTD, eNAC, NAC-Sarsa, and INAC, follow *biased* estimates of the natural policy gradient [29]. For our experiments, we must use an unbiased algorithm since the projection that we propose is compatible with the natural gradient, but not necessarily biased estimates thereof.

NAC-Sarsa and INAC are equivalent *biased* discounted-reward natural actor-critic algorithms with per-time-step time complexity linear in the number of features. The former was derived by replacing the LSTD-Q($\lambda$) component of NAC-LSTD with Sarsa($\lambda$), while the latter is the discounted-reward version of NGAC. Both are similar to NTD, which is a biased average-reward algorithm. The *unbiased discounted*-reward form of NAC-Sarsa was recently derived [29]. References to NAC-Sarsa hereafter refer to this unbiased variant. In our case studies we use the *projected natural actor-critic using Sarsa($\lambda$)* (PNAC-Sarsa), the projected version of the unbiased NAC-Sarsa algorithm.

Notice that the projection, $\Pi_\Theta^{G(\theta_k)}$, as defined in (5), is *not* merely the Euclidean projection back onto $\Theta$. For example, if $\Theta$ is the set of $\theta$ that satisfy $A\theta \leq b$, for some fixed matrix $A$ and vector $b$, then the projection, $\Pi_\Theta^{G(\theta_k)}$, of $y$ onto $\Theta$ is a quadratic program,

$$\text{minimize } f(\theta) = -y^\mathsf{T} G(\theta_k)\theta + \frac{1}{2}\theta^\mathsf{T} G(\theta_k)\theta, \qquad \text{s.t. } A\theta \leq b.$$

In order to perform this projection, we require an estimate of the average Fisher information matrix, $G(\theta_k)$. If the natural actor-critic algorithm does not already include this (like NAC-LSTD and NAC-Sarsa do not), then an estimate can be generated by selecting $G_0 = \beta I$, where $\beta$ is a positive scalar and $I$ is the identity matrix, and then updating the estimate with

$$G_{t+1} = (1 - \mu_t)G_t + \mu_t \left( \frac{\partial}{\partial \theta_k} \log \pi(a_t|s_t, \theta_k) \right) \left( \frac{\partial}{\partial \theta_k} \log \pi(a_t|s_t, \theta_k) \right)^\mathsf{T},$$

where $\{\mu_t\}$ is a step size schedule [14]. Notice that we use $t$ and $k$ subscripts since many time steps of the MDP may pass between updates to the policy parameters.

## 10   Case Study: Functional Electrical Stimulation

In this case study, we searched for proportional-derivative (PD) gains to control a simulated human arm undergoing FES. We used the Dynamic Arm Simulator 1 (DAS1) [30], a detailed biomechanical simulation of a human arm undergoing functional electrical stimulation. In a previous study, a controller created using DAS1 performed well on an actual human subject undergoing FES, although it required some additional tuning in order to cope with biceps spasticity [6]. This suggests that it is a reasonably accurate model of an ideal arm.

The DAS1 model, depicted in Figure 2a, has state $s_t = (\phi_1, \phi_2, \dot{\phi}_1, \dot{\phi}_2, \phi_1^{target}, \phi_2^{target})$, where $\phi_1^{target}$ and $\phi_2^{target}$ are the desired joint angles, and the desired joint angle velocities are zero. The goal is to, during a two-second episode, move the arm from its random initial state to a randomly chosen stationary target. The arm is controlled by providing a stimulation in the interval $[0, 1]$ to each of six muscles. The reward function used was similar to that of Jagodnik and van den Bogert [6], which punishes joint angle error and high muscle stimulation. We searched for locally optimal PD gains using PNAC-Sarsa where the policy was a PD controller with Gaussian noise added for exploration.

Although DAS1 does not model shoulder dislocation, we added safety constraints by limiting the $l_1$-norm of certain pairs of gains. The constraints were selected to limit the forces applied to the humerus. These constraints can be expressed in the form $A\theta \leq b$, where $A$ is a matrix, $b$ is a vector, and $\theta$ are the PD gains (policy parameters). We compared the performance of three algorithms:

   1. **NAC**: NAC-Sarsa with no constraints on $\theta$.

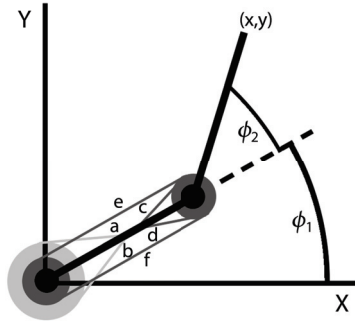

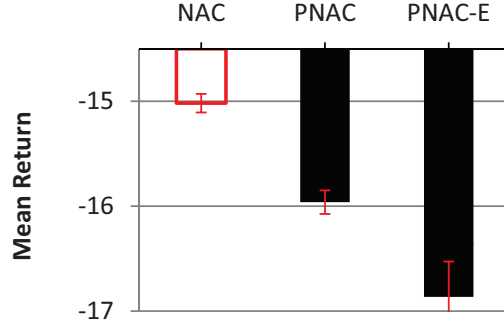

(Figure 2a) DAS1, the two-joint, six-muscle biomechanical model used. Antagonistic muscle pairs are as follows, listed as (flexor, extensor): monoarticular shoulder muscles (a: anterior deltoid, b: posterior deltoid); monoarticular elbow muscles (c: brachialis, d: triceps brachii (short head)); biarticular muscles (e: biceps brachii, f: triceps brachii (long head)).

(Figure 2b) Mean return during the last 250,000 episodes of training using thee algorithms. Standard deviation error bars from the 10 trials are provided. The NAC bar is red to emphasize that the final policy found by NAC resides in the dangerous region of policy space.

2. **PNAC**: PNAC-Sarsa using the compatible projection, $\Pi_{\Theta}^{G(\theta_k)}$.
3. **PNAC-E**: PNAC-Sarsa using the Euclidean projection.

Since we are not promoting the use of one natural actor-critic over another, we did not focus on finely tuning the natural actor-critic nor comparing the learning speeds of different natural actor-critics. Rather, we show the importance of the proper projection by allowing PNAC-Sarsa to run for a million episodes (far longer than required for convergence), after which we plot the mean sum of rewards during the last quarter million episodes. Each algorithm was run ten times, and the results averaged and plotted in Figure 2b. Notice that PNAC performs worse than the unconstrained NAC. This happens because NAC leaves the safe region of policy space during its search, and converges to a dangerous policy—one that reaches the goal quickly and with low total muscle force, but which can cause large, short, spikes in muscle forces surrounding the shoulder, which violates our safety constraints. We suspect that PNAC converges to a near-optimal policy within the region of policy space that we have designated as safe. PNAC-E converges to a policy that is worse than that found by PNAC because it uses an incompatible projection.

## 11 Case Study: uBot Balancing

In the previous case study, the optimal policy lay outside the designated safe region of policy space (this is common when a single failure is so costly that adding a penalty to the reward function for failure is impractical, since a single failure is unacceptable). We present a second case study in which the optimal policy lies within the designated safe region of policy space, but where an unconstrained search algorithm may enter the unsafe region during its search of policy space (at which point large negative rewards return it to the safe region).

The uBot-5, shown in Figure 3, is an 11-DoF mobile manipulator developed at the University of Massachusetts Amherst [31, 32]. During experiments, it often uses its arms to interact with the world. Here, we consider the problem faced by the controller tasked with keeping the robot balanced during such experiments. To allow for results that are easy to visualize in 2D, we use a PD controller that observes only the current body angle, its time derivative, and the target angle (always vertical). This results in the PD controller having only two gains (tunable policy parameters). We use a crude simulation of the uBot-5 with random upper-body movements, and search for the PD gains that minimize a weighted combination of the energy used and the mean angle error (distance from vertical).

We constructed a set of conservative estimates of the region of stable gains, with which the uBot-5 should never fall, and used PNAC-Sarsa and NAC-Sarsa to search for the optimal gains. Each training episode lasted 20 seconds, but was terminated early (with a large penalty) if the uBot-5 fell over. Figure 3 (middle) shows performance over 100 training episodes. Using NAC-Sarsa, the PD weights often left the conservative estimate of the safe region, which resulted in the uBot-5 falling over. Figure 3 (right) shows one trial where the uBot-5 fell over four times (circled in red). The

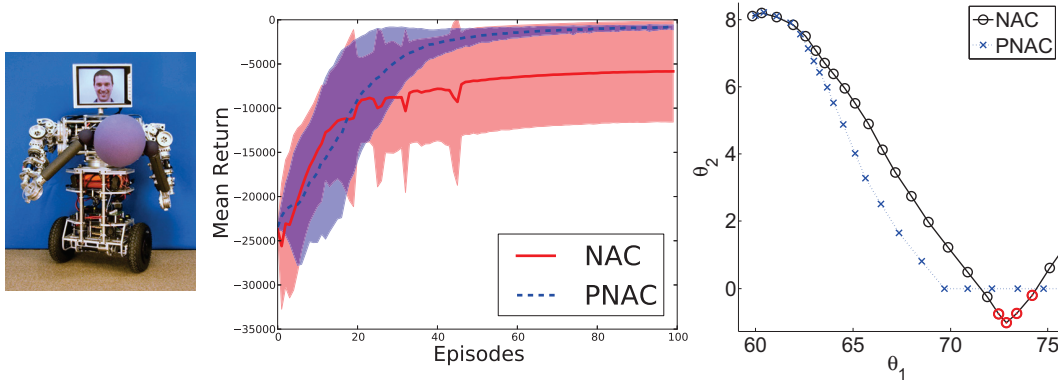

Figure 3: **Left:** uBot-5 holding a ball. **Middle:** Mean (over 20-trials) returns over time using PNAC-Sarsa and NAC-Sarsa on the simulated uBot-5 balancing task. The shaded region depicts standard deviations. **Right:** Trace of the two PD gains, $\theta_1$ and $\theta_2$, from a typical run of PNAC-Sarsa and NAC-Sarsa. A marker is placed for the gains after each episode, and red markers denote episodes where the simulated uBot-5 fell over.

resulting large punishments cause NAC-Sarsa to quickly return to the safe region of policy space. Using PNAC-Sarsa, the simulated uBot-5 never fell. Both algorithms converge to gains that reside within the safe region of policy space. We selected this example because it shows how, even if the optimal solution resides within the safe region of policy space (unlike the in the previous case study), unconstrained RL algorithms may traverse unsafe regions of policy space during their search.

## 12   Conclusion

We presented a class of algorithms, which we call *projected natural actor-critics* (PNACs). PNACs are the simple modification of existing natural actor-critic algorithms to include a projection of newly computed policy parameters back onto an allowed set of policy parameters (e.g., those of policies that are known to be safe). We argued that a principled projection is the one that results from viewing natural gradient descent, which is an *unconstrained* algorithm, as a special case of mirror descent, which is a *constrained* algorithm.

We show that the resulting projection is compatible with the natural gradient and gave a simple empirical example that shows why a compatible projection is important. This example also shows how an incompatible projection can result in natural gradient descent converging to a pessimal solution in situations where a compatible projection results in convergence to an optimal solution. We then applied a PNAC algorithm to a realistic constrained control problem with six-dimensional continuous states and actions. Our results support our claim that the use of an incompatible projection can result in convergence to inferior policies. Finally, we applied PNAC to a simulated robot and showed its substantial benefits over unconstrained natural actor-critic algorithms.

## Footnotes

[1]Later, we abuse notation and switch freely between treating $G_k$ as a matrix and a relation. When it is a matrix, $G_k x$ denotes matrix-vector multiplication that produces a vector. When it is a relation, $G_k(x)$ produces the singleton $\{G_k x\}$.

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
