[Reviews · NeurIPS 2013]

Submitted by Assigned_Reviewer_7

Projected Natural Actor-Critic

Summary:
The authors propose a version of the Natural Actor-Critic policy search algorithm that can be constrained in its search space. Thus, it can be used for, e.g., medical tasks where some regions of the action space might be dangerous to execute.
The authors start by showing that natural gradient descent is a special form of mirror descent and use that insight to formulate a version of mirror descent that includes the safety guarantees. These guarantees can alternatively be used to include domain knowledge in the learning process. The algorithm guarantees to select optimal steps even under the constraints.


Quality:
The overall quality of the paper is good. The method proposed by the paper is derived cleanly.
The incorporation of domain knowledge seems interesting, but I am not sure how easy it is to describe this knowledge for more complex systems and whether for such systems a tunnel of possible actions derived from demonstrations might not be a better approach. It would be interesting to read about this in the discussion section. I am also wondering how the algorithm would compare to a naive approach of manually rejecting and resampling dangerous actions.
The experimental section of the paper is somewhat weak, an evaluation on more complex systems and tasks would be welcome.


Clarity:
The paper is well written. The authors take a clear path to derive their method and explain the individual steps and insights well.


Originality:
As the authors state, the idea of constraining the actions of a RL method itself is not new, however, the integration into the actor-critic method itself is new.


Significance:
The paper is a significant contribution to the community. However, I am wondering of the applicability outside of the medical regime.
Summary: The submission is a valuable contribution to the field of policy search and well written. The experimental section of the paper is somewhat weak.

Submitted by Assigned_Reviewer_8

Summary
The paper is about the proposal of a class of constrained natural actor critics, where, for safety reasons, policy parameters must remain in a subregion.
The idea is to apply natural actor critic algorithms, that update policy parameters by following the estimated direction of the natural policy gradient and, whenever the policy parameters get out of the safe region, the parameters are projected back to allowed values.
The authors show that natural gradient ascent is a particular case of mirror ascent, and, being the latter a constrained optimization algorithm, the projection can be simply (and effectively) obtained by adding constraints to the policy parameters values. Besides theoretically proving that the resulting projection is compatible with the natural policy gradient, a simple example and two more complex case studies have been introduced to evaluate the performance of the proposed solution and the negative effects that can derive in critical systems when either unconstrained optimization or a wrong projection method are used.

Quality
The paper is technically sound. The theoretical claims are correctly proved and the effectiveness of the proposed approach is empirically evaluated in medium-complexity domains.
The authors have made a good work in highlighting the advantages of the proposed approach, while they have not mentioned to potential weakness and no future research direction has been proposed.
For instance, I would like to read some discussion about the complexity of computing the projection and to see some example where the constraints over the parameters are not linear.

Clarity
The paper is well written. The paper is structured into many sections. I suggest to reorganize the paper into a smaller number of sections that collects the current sections as subsection.
For instance, I would group sections 3,4,5, sections 6,7, sections 8,9, and section 10,11. Furthermore, I would move the related work section at the end, before the conclusions.
I found particularly instructing the example of Section 7, that allows the reader to clearly visualize how the proposed approach moves in the parameter space and what may happen by following other (naive) approaches.
For what concerns the reproducibility of experimental results, the example in Section 7 can be easily reproduced, while for the two case studies some details are missing about the domain.
In general, no information have been reported about the value of the step size used to update the parameters along the gradient direction.

Originality
As far as I know, the proposed approach is novel. In the related work section, the authors cite other approaches to problems similar to the one faced in this paper.
As the authors state in the conclusions, the contribution is a "simple" modification to standard natural actor-critic algorithms to include projection.
Such modification is simple since, by choosing the correct proximal function, natural gradient ascent can be realized through the constrained optimization carried out by mirror descent.

Significance
The setting faced by this paper is really relevant.
In the last years, policy search and, in particular, policy gradient approaches have been largely used to learn policy for complex continuous tasks.
On the other hand, perform an unconstrained optimization in the policy space may be dangerous.
This paper advances the state of the art by introducing a principled way to constrain the search for one prominent class of policy gradient algorithms: natural actor critic algorithms.
I think that this line of research is interesting for many researchers (especially in the field of robotics) and this paper can have a positive impact on future researches.
Summary: The approach proposed in this paper for constraining the search performed by natural actor-critic algorithms to safe policies is theoretically sound and empirically effective.
A good paper that would benefit from some additional discussion about the weakness of the proposed approach.

Submitted by Assigned_Reviewer_9

The paper addresses the problem of constraining the parameters of the policy to be within a set of valid, or safe, parameters. This is a very important problem in policy gradient methods, wherein the gradient updates may lead to arbitrary policies that can be unsafe to execute. The authors start by first showing that the natural policy gradient method is a special case of the more general mirror descent method with a proximal function corresponding to the norm using the manifold metric (e.g. the Fischer information matrix). The authors then derive the projected natural gradient by adding the domain constraints to the conjugate of the proximal function in the mirror descent. Finally, the authors apply the new projection method to natural actor-critic algorithms. The resulting algorithm is quite simple: For each step, perform a natural gradient update then find the closest point in the set of valid parameters using the metric of the manifold instead of the Euclidean metric. The authors demonstrate the safety of the policies obtained by their projection algorithm on an arm control problem, and a ball balancing problem.

Quality:
The derivation of the projection algorithm seems sound and correct. The empirical evaluation is somehow limited, but still shows the advantage of the proposed algorithm. The paper also unifies the natural gradient descent algorithm with the mirror descent algorithm.

Clarity:
The paper is generally clear, except in a key part that is Equation (4), using the same notations for relations and functions, and switching between them was not clear to me. Also, a function that returns infinity at a given point is not a familiar concept, how can you derive it? The authors should explain these points better.

Originality:
This work is novel, up to my knowledge. However, the relation between natural gradient and mirror descent seems fairly trivial, I am not sure if nobody noticed it before. Moreover, the final projection algorithm with natural gradient seems trivial. The only difference with other algorithms is the use of the manifold metric for the projection. The authors should also take a look to the Relative Entropy Policy Search (REPS) algorithm that solves the same problem of bounding the policy parameters with a quite similar approach, by bounding the KL divergence with respect to a baseline policy, using thus the tensor metric (Fischer information matrix) for constraining the parameters instead of the Euclidean one. However, the proposed method is more general and can include any type of policy constraints.

Significance:
This work is significant in the sense that it addresses an important problem, and derives the solution in a principled way.
Summary: The paper presents a nice derivation of a new gradient projection algorithm, and a unified view of natural gradient and mirror descent. I would like to see the authors discuss the relation to the Relative Entropy Policy Search, which seems closely related.
Author Feedback

Author rebuttal: Thank you for your feedback. It will help to improve the clarity and completeness of the paper.